# Acceptability of a Smartphone Application to Enhance Healthcare to Female Genital Mutilation Survivors in Liberia: A Qualitative Study

**DOI:** 10.3390/ijerph191710855

**Published:** 2022-08-31

**Authors:** Kim Nordmann, Guillermo Z. Martínez-Pérez, Mandella King, Thomas Küpper, Ana Belén Subirón-Valera

**Affiliations:** 1Institute of Occupational & Social Medicine, RWTH Aachen University, 52062 Aachen, Germany; 2African Women’s Research Observatory, 08009 Barcelona, Spain; 3Saint Joseph’s Catholic Hospital, Monrovia 1000, Liberia; 4Department of Physiatrics and Nursing, University of Zaragoza, 50009 Zaragoza, Spain; 5Research Group Water and Environmental Health (B43_20R), University Institute of Research in Environmental Science of Aragón, University of Zaragoza, 50009 Zaragoza, Spain; 6Research Group Sector III Healthcare (GIIS081), Institute of Research of Aragón, 50009 Zaragoza, Spain

**Keywords:** Liberia, mHealth, female genital mutilation, knowledge acquisition, self-directed learning

## Abstract

In Liberia, female genital mutilation/cutting (FGM/C) is a legally allowed initiation ritual in the secret Sande society. Due to the secrecy, Liberian healthcare providers receive little education on FGM/C and its health consequences. As mobile learning approaches proved to efficiently increase providers’ knowledge and skills, a mobile application (‘app’) was designed to support self-learning, decision-making, and the follow-up of FGM/C survivors’ health. The ‘app’ was introduced in a capacity-building project in 2019 and evaluated through this qualitative study to assess healthcare provider’s needs and acceptance. We conducted 22 semi-structured interviews and eight focus group discussions with 42 adult healthcare providers in three Liberian counties. A thematic approach grounded in descriptive phenomenology guided data analysis and led to three main themes: the ‘app’, mobile learning and health education, and personal impression. Healthcare providers judge the ‘app’ useful to broaden their knowledge and skills, which might lead to better FGM/C detection and management. The ‘app’ might further facilitate patient and community education about the negative health consequences of FMG/C, possibly contributing to a reduction of FGM/C prevalence.

## 1. Introduction

Female genital mutilation/cutting (FGM/C) is one of the most harmful traditional genital modifications for non-medical purposes on girls and women. It is grouped into four types (Table 1) and may result in immediate and long-term negative health outcomes, such as infertility and an increased risk of obstetric complications [1]. Most FGM/C-practicing countries already banned FGM/C [2]. However, even though Liberia ratified the Maputo Protocol—an additional protocol to the African Charter on Human and Peoples’ Rights focusing on the rights of women in Africa and explicitly stating the prohibition of all forms of FGM/C—Liberia is among the few countries that are yet to enforce regulations to end the practice of FGM/C [3]. Attempts to introduce an FGM/C-abolishment clause in the Domestic Violence Law in 2016 were unsuccessful, and a one-year executive order (No. 92) in 2018 prohibiting FGM/C was not extended [4,5,6].

In Liberia, FGM/C is performed during an initiation ritual to the Sande society. The Sande society is a female-only and ancient society that holds significant social power in Liberia. Due to the secrecy around the Sande society, there is little room for public debate about FGM/C in Liberia [7]. In-country FGM/C-ban opposition from the Sande society is among the reasons why 38.2% of women 15–49 years old are still living with the consequences of FGM/C [8]. This secrecy and the lack of institutional support for its eradication hinders advocacy and educational activities on the negative health consequences of FGM/C [7].

The training of the local healthcare workforce on sensitizing practicing communities is a strategy to eradicate FGM/C [9,10,11]. Additionally, FGM/C survivors benefit from healthcare providers trained on the management of long-term obstetric, sexual and mental consequences [9,12,13]. The World Health Organization (WHO) has therefore released guidelines to inform healthcare providers about evidence-based care to FGM/C survivors [14]. While face-to-face training on WHO guidelines needs significant resources, mobile learning (mLearning) is a feasible option. There is evidence that mLearning interventions, such as smartphone applications or web-based toolkits, are cost-efficient and can increase providers’ knowledge and skills [15]. A 2020 review identified 31 web-based toolkits with information about FGM/C and medical treatment recommendations targeting healthcare providers [16]. There exists one mLearning FGM/C application for healthcare providers in the United Kingdom [17]. To our knowledge, no mLearning mobile phone FGM/C application has been proposed for the Liberian context.

A mobile application (hereafter, ‘app’) was proposed to enhance the obstetric and psychosocial skills of Liberian healthcare providers to care for FGM/C survivors. Framed as a cooperation-for-development action of the acronym YOUCANTRY!, healthcare providers received the ‘app’ as a supplementary tool when attending a 2-day workshop on FMG/C and its health consequences. The workshops, carried out in 2019, were designed based on a 2017 study on midwives’ experiences providing obstetric care to FGM/C survivors in Liberia [7]. Ancillary to YOUCANTRY!, we conducted qualitative research to assess the acceptability of the FMG/C ‘app’ and the needs of the Liberian healthcare providers concerning the ‘app’.

## 2. Materials and Methods

### 2.1. Setting and Design

This qualitative inquiry was conducted in the Liberian counties of Montserrado, Nimba and Lofa between February and September 2019, alongside the capacity-building activities included in the project YOUCANTRY!. Within YOUCANTRY!, a series of six workshops aimed to improve the professional competence of nurses, midwives, physician assistants and traditional midwives (hereafter, HCP) to identify and manage FGM/C-related complications. The workshops were guided by the WHO’s handbook ‘Care of girls and women living with female genital mutilation’ [14] and were attended by a total of 133 HCPs. The results of the effectiveness evaluation of the workshops are published elsewhere [18].

An ‘app’ to support trainees’ knowledge acquisition was introduced in some workshops and tested with consenting participants during the qualitative research. The ‘app’ was developed using the open-source data collection tool KoBoToolbox^®^. The ‘app’s purpose was to self-learn about FGM/C management and to self-assess the adequacy of clinical decisions with FGM/C survivors. It included a series of decision-aid algorithms based on the handbook ‘Care of girls and women living with female genital mutilation’ [14].

### 2.2. Population and Sampling Procedure

The population of this research were (i) adult HCPs attending the workshops and (ii) their adult workmates. Eligible participants had to be engaged in healthcare provision to FGM/C survivors and willing to provide informed consent.

K.N. (a white and female German medical school student) explained the purpose of the qualitative research at the beginning of each workshop and informed attendees that they could be approached in person or by phone. After each workshop, the attendees were contacted and invited to participate in the research. They were also asked to recommend workmates meeting the study inclusion criteria. A venue and date for data collection was accorded for those interested in the study. Purposive sampling continued alongside the workshops until the project YOUCANTRY! concluded. In total, 27 women (mean age 42.2 years) and 15 men (mean age 34.5 years) participated in the study (Table 2). Two approached individuals refused to participate, and no reason was provided. Five midwives, 15 nurses, six physician assistants and twelve trained traditional midwives participated, of which two were both a nurse and a midwife. Most participants were born in the counties where the workshops took place.

### 2.3. Data Collection and Analysis

Prior to data collection, informed consent from all participants was sought. Consenting participants were invited to partake in semi-structured individual interviews (SSIs) and focus group discussions (FGDs). These data collection methods allow for the rich and culturally grounded comprehension of people’s experiences [19].

Data collection was conducted in English at a place of participants’ choice, who mostly opted for their workplace in case of SSI or the workshop venue for FGD. At the start of the SSI and FGD, participants provided sociodemographic, as well as mobile ownership and utilization, data. A total of 22 SSIs (average 64 min) and eight FGDs (average 59 min) were conducted. Participation in the FGDs ranged from two to six individuals. Six participants took part in several data collection activities.

All SSIs and FGDs followed the same semi-structured interview guide, in which emerging themes were constantly incorporated. All SSIs and all but one FGD were tape-recorded. From the non-recorded FGD, notes were taken.

Audio was transcribed with easytranscript (E-Werkzeug, Berlin, Germany) into Microsoft Word^®^ (Microsoft Corporation, Redmond, WA, USA) and stored on a password-protected computer. Transcriptions were cross-checked against the recordings, amended in case of inconsistencies, and uploaded to Nvivo V.12^®^ (RRID:SCR_014802, QSR International, Burlington, MA, USA). All transcripts were inductively coded in Nvivo V.12^®^ (RRID:SCR_014802) using thematic analysis as described by Braun and Clark [20]. This approach was selected as it allows understanding complex phenomena—such as healthcare professionals’ perceptions of the ‘app’—through the study populations’ lived experiences and perspectives [21].

Constant comparative analysis was applied to ensure the trustworthiness of the data analysis. The transcripts were analyzed by two researchers (K.N., A.S.). Themes and subthemes were compared. Participants’ narratives were triangulated with field notes taken during data collection and with participant observation reports of the workshops and other workshop-related data. The researcher leading field data generation (K.N.) practiced memoing to identify the potential influences of her influence on the participants’ viewpoints and narratives. The possible effect of memory, social desirability and observant biases was discussed.

The reporting of the findings in this article considers the consolidated criteria for reporting qualitative studies (COREQ) checklist [22].

### 2.4. Ethics

Participants consented after receiving detailed information on the study. They received information about the study aim, the risk of social harm that may derive from their participation, the data management procedures to protect their identities and confidentiality of their information, and their right to skip any question at any time or to withdraw from the study without any negative consequences. In case the participants were illiterate, the information sheet was read to them. A signed copy of the information sheet and consent form remained with the participant.

Reimbursement of the transport expenses was offered to the participants. No payment for participation in the research was given.

All participants received a unique identification number, which was the only identifier linking the consent forms to the transcript and socio-demographic data. All consent forms were safely stored in the principal’s investigators office. Audio recordings were destroyed after coding the transcriptions to ensure confidentiality.

The University of Liberia-Pacific Institute for Research Evaluation Institutional Review Board approved the study protocol (Ref. #19-01-148).

## 3. Results

In this section, study findings are presented alongside three core themes: the ‘app’, mobile learning and personal impressions. Words, terms and expressions used by the participants are in *italics*. Unless indicated otherwise, the acronym HCP is used where insights and narrations were commonly shared by all cadres of HCPs involved as participants (i.e., nurses, midwives, physician assistants, trained traditional midwives).

### 3.1. The ‘App’

#### 3.1.1. Utility

HCPs expressed that they wanted to use the ‘app’ for *personal study* to keep and *build up knowledge*. Some participants would like to study together with colleagues and discuss the content with peers. The ‘app’ might be used as a tool to inform non-workshop participants about its content and the WHO guidelines on obstetric, sexual and mental health care provision for FGM/C survivors. As a male nurse detailed:


*“My major expectation is to be able to help me and guide me when I am using it practically. So that I can get at a result that will benefit my patient.”*
(10, SSI)

When treating FGM/C survivors, HCPs are confronted with secrecy around FMG/C. Most HCPs mentioned repercussions and danger to their lives if they openly talked about FGM/C. Therefore, HCPs imagined using the ‘app’ as a tool that could legitimize their talking about FGM/C in their clinics. A variation of media resources, above all pictures and videos, gave credibility to the words of the HCPs as *seeing is believing*.


*“They will see the pictures because the rural people believe in pictures. Seeing is believing. Because if you just go and talk to them like the way you talk to us, without seeing no pictures, no examples, they will not be convinced easily. But wherein you show them the good side and the negative side, and they see it, they can analyse for themselves and make decision and decide.”*
(30, FGD)

#### 3.1.2. Understandability

The ‘app’ should be directed at midwives, nurses and physician assistants as front-line workers. Generally, participants suggested presenting content in high-school-level English. However, if the ‘app’ was to be used by TTM, it should be adapted—as hinted by a female midwife—*in a way that they will understand*.

HCPs recommended inviting future ‘app’ users to trainings on FGM/C. Due to the secrecy around FMG/C and the lack of education during the paramedical formation, training could improve the understanding of the ‘app’. As one female TTM pointed out:


*“Reading is one thing and understanding is another one.”*
(18, SSI)

#### 3.1.3. Usability

Those HCPs favoring the ‘app’ over the use of printed guidelines valued the possibility of updates and the availability of the guidelines as compared to paper-based guidelines, which might be locked in the health centre’s office.


*“So, this one [an app] is make it easy, anywhere you go, you take it, you study along with it.”*
(11, SSI)

HCPs recommended including a cultural section in the ‘app’ to raise HCPs' awareness on the beneficial and harmful effects of Liberian cultural practices such as FGM/C on health. Some participants pointed out that the ‘app’ should respect *African culture* at all points, i.e., be culturally sensitive when talking about cultural practices and their consequences and display to its user that efforts to reduce the prevalence of FGM/C are not limited to Liberia.


*“So, when they have seen it [the video] themselves, they will know that it is not only in Liberia that people are against it [FGM/C] or that people are trying to eradicate it. […] You tell them that ‘We are not against it but the health benefits—the side effects of it—that is what we are discussing’.”*
(33, SSI)

Some participants advised including information on complications in maternal and mental health beyond FGM/C. A male nurse highlighted that the *de-traumatization process* in post-war Liberia was lacking and that many patients were still suffering from its consequences, direct consequences as veterans and indirect consequences through the behavior of traumatized war survivors. He detailed:


*“The war in Liberia was protracted. It was bitter. It was long. So, we think that, for me—I think that it’s because of that people are developing those conditions [anxiety and depression]. […] if you look at the process of de-traumatization that they carry on after the war, to me, as a person, I don’t think it was enough. I think they did it hastily.”*
(19, SSI)

Next to obstetric complications and its management, HCPs suggested incorporating education about mental health counseling. A female registered nurse and mental health clinician recommended adding themes such as confidentiality and *rapport-building*. A male physician assistant further explained:


*“Some counselling methods should be provided in the sense that based on the case presenting either be it familiar or be it some stress condition, socially or because of the result of you being mutilated that you cannot socially be accepted, you know, some guide of counselling should be provided.”*
(41, FGD)

Some participants suggested integrating sexual health topics, i.e., approaches to common sexual problems and misconceptions about sexual health and strategies how to begin conversations about sexual health with the patients.


*“So, there should be more information on sexual health in order to correct some of those problems.”*
(42, FGD)

#### 3.1.4. Feasibility

Smartphone ownership was at 69% in the respondent group (Table 3). Participants reported that they shared phones, although they described it as a source of trouble. A male nurse recalled that using his colleague’s phone was *embarrassing* to his colleague, and he therefore bought his own phone.

Some HCPs frequented a charging booth due to a lack of electricity in their houses and limited security when charging it in the clinic. A few HCPs were concerned that smartphones might be stolen or break and that a lack of money might impede smartphone (re)uptake and ‘app’ usage. As a female nurse stated:


*“This phone any time can spoil, or someone can steal it.”*
(13, SSI)

Most HCPs used apps, such as Facebook and Messenger, *to socialize,* and some HCPs used them to play games or to consult the Bible. Apps for clinical practice were frequently used (57.1%): HCPs consulted applications to look up medical terms, guidelines and medication dosage, to study and to cross-check symptoms and diseases. Some HCPs checked the quality of information obtained from the clinical app by comparing it to a book or internet pages. The phone was also used for calling—not videoconferencing—doctors and colleagues if problems were encountered or for taking pictures of certain conditions.


*“I use my smartphone for confirmation when it comes to drug interaction, drug mechanism of action, drug contraindication and maybe side-effect. And one more thing is that I use my phone for additional information as it relates to clinical signs and symptoms of a case to conclude with a diagnosis.”*
(41, FGD)

As some HCPs encountered problems installing the ‘app’ due to the limited storage capacity of their smartphones, a small app size was preferable. HCPs recommended that the ‘app’ be available offline and free, making a transfer of the ‘app’ through Xender/CShare between HCPs possible. Despite concerns about the insecurity of smartphone ownership and charging and difficulties installing the ‘app’, some HCPs manifested interest in the ‘app’, while others preferred the WHO guidelines as a paper-based poster or booklet. Training on the ‘app’ might enhance its usage by HCPs who are less mobile literate.


*“For the app, since I’m just getting to learn about this phone, as I said, I have a little bit problem with it but I’m quite sure it says practice makes perfect.”*
(21, SSI)

HCPs were ambivalent about using the ‘app’ in front of the patient. While some argue that patients would accept it after an explanation and even perceive the ‘app’ as more confidential than anything paper-based, a male nurse highlighted patients’ fears that the HCPs could share confidential information with the public. Further, consulting the ‘app’ in front of the patient might decrease patients’ trust in HCPs’ skills.


*“Some of the patients will discredit you. You’re on the phone. [...] They will say ‘you don’t know what you are doing’.”*
(5, SSI)

### 3.2. Mobile Learning and Health Education

#### 3.2.1. Knowledge about FGM/C

Most participants reported that they received either no or little education about FMG/C in their paramedical formation. A male registered nurse mentioned that HCPs knowledgeable on FGM/C were educated outside of Liberia. Certified midwives seemed to have obtained more training on FMG/C than other educational groups. One certified midwife self-trained on FGM/C. Albeit the lack of training, HCPs expressed being able to recognize FMG/C. However, one HCP reported not having talked about FMG/C with the patients out of fear of saying the wrong thing.


*“First, we never knew the importance of this. We never knew that it was not good for women. We never knew the side-effects. But nowadays, we’re learning day by day, when we’re seeing the side-effects of FGM.”*
(23, FGD)

HCPs advised to include education about FGM/C in college and university curricula for certified midwifes, registered nurses and physician assistants. A male nurse advocated for training traditional midwifes on FGM/C, since they are *with the people day and night*.

Respondents perceived the ‘app’ as a resource for self-learning on the complications of FGM/C. Short bullet point messages and multimedia support through educational videos might increase HCPs' understanding and *reinforce* what was learned in the workshops. Videos might teach practical skills, such as performing a genital assessment of a patient with FMG/C, or the emotional skills to approach an FMG/C patient.

In their opinion, pictures and illustrational videos would further enable HCPs to educate survivors about the management of existing or future complications, since not all patients are literate and since many *learn visually*. A female nurse explained that pictures comparing a vagina with FMG/C against a vagina without FMG/C might help to begin the conversation. HCPs expected that such an mLearning resource would help them to convince patients to be referred to a higher-level facility.


*“Pictures can have a lasting, lasting influence on the mind of people. When they see pictures of something, it remains, it sticks like it sticks on their minds.”*
(20, SSI)

An app directed at HCPs might use high-school-level English. Should the ‘app’ include tools to promote interaction between an HCP and a survivor, the language needs to be adapted depending on the attitude of the patient towards FGM/C; respectfully talking about the society, while pointing out the negative health effects, as a talk against FMG/C might be interpreted as a talk against the Sande society itself. HCPs recommended conducting an interactive quiz with the patient after the health education to ensure the patient’s understanding of the teaching. HCPs would adapt the language used in their health education to the needs of each patient—paraphrasing pregnant woman with *big belly*, urine with *pee-pee* and sex with *meet your husband/lay down inside with your husband*.


*“I’m not going to talk with native woman, I will be using big, big book [extensive vocabulary] […] She will not get anything. So, I need to break it […] down to her level and even speak it in Liberian way.”*
(23, FGD)

#### 3.2.2. Management of FGM/C

Many HCPs expressed not having focused on FGM/C during their clinical practice and did not correlate FGM/C to the observed complications, interpreting them to be *normal pregnancy complications*. Only a few HCPs associated FGM/C with a higher risk of health complications and have given health education about FGM/C to patients before. The majority stated that they did not offer counseling or health education to FGM/C survivors in their consultations.


*“If complications happened, we just think that it’s medical problem. We were not thinking about FMG [FGM/C].”*
(34, SSI)

One HCP shared her approach to avoid lacerations in FGM/C survivors. HCPs of primary care facilities usually referred FGM/C patients due to a lack of doctors and operation room. One HCP explicitly mentioned that primipara FGM/C survivors are usually referred, while multipara FGM/C survivors with no other risk factor might give birth at the facility.


*“Well, when I come across this kind of case I would like to take it to the hospital instead of treating at myself.”*
(21, SSI)

HCPs reported that opening the conversation and talking about FGM/C would be *challenging*. They requested that ideas on how to discuss FMG/C with their patients be incorporated in the ‘app’, focusing on the timing and strategy of raising the topic. Furthermore, they mentioned the open attitude the HCP should portray: *sympathizing with the patient on their situation* and treating FGM/C survivors with respect. The ‘app’ should provide ideas to *reason* with *traditional people* [i.e., people who belong to the Sande society] who might be in favor of FGM/C.


*“Because some people if they come from down there, they are very hard to reason. We talk, talk, talk, talk, talk. They will say ‘that is our tradition thing’. So, I would love to have a few guidelines.”*
(23, FGD)

#### 3.2.3. Acceptability of the WHO Guidelines

HCPs welcomed the guidelines on the management of FGM/C survivors and were willing to implement them in their clinical practice. This included marking the patient’s FGM/C status in the patient record to inform their colleagues about the higher risk of complications and changes in screening and history-taking. Vaginal assessment in antenatal care was in many HCPs' workplaces not routinely done, and, to some, it seemed difficult to implement in their assessment. One HCP pointed out that the uptake of the guidelines depended on the FGM/C prevalence of the region, with regions of lower FGM/C prevalence having a quicker uptake. A male physician assistant requested the information displayed in the ‘app’ to be in-line with the WHO or national guidelines. Due to secrecy and taboo around sexuality, and especially in counties where the Sande society was widespread, the WHO guidance on sexual health seemed to be more difficult to implement than guidance on mental or obstetric health. A female registered nurse expressed that conversations about sex between a couple were not the norm:


*“Sometime your dislike, you have to express it. But for our setting it is hard for women to express their dislike, especially when it comes to sex.”*
(34, SSI)

### 3.3. Personal Impression

#### 3.3.1. Experiences with FGM/C Survivors

HCPs emphasized that talking about FMG/C was a taboo, hindering an open conversation in clinical practice. Commonly, patients considered FGM/C a secret and preferred to be treated by a clinician who was a member of the Sande society. One HCP stated that raising the topic of FGM/C was easier with younger patients, as older patients might feel *insulted*. To start talking about FGM/C, several HCPs envisioned a one-to-one conversation with patients but requested prior *authorization* from the government, e.g., through an anti-FGM/C law.


*“We have FGM is a topic but as I said it is like taboo. So, you have to be very careful of how you explain.”*
(19, SSI)

HCPs noted that FGM/C survivors felt *ashamed* of their condition, not wanting to *expose* themselves (i.e., remove their clothes for examination) in front of the HCP or to talk about FGM/C.


*“This condition is something that people can be ashamed of. They can’t be easily release themselves to talk about it.”*
(18, SSI)

Several HCPs affirmed that FGM/C type I is the most common type practiced in Liberia. A female nurse and midwife recalled FGM/C victims with a variation of type I, wherein only one of the labia minora was cut. One HCP reported having seen type III on a woman from Guinea. Sexual abstinence, belonging to a group and sharing a group identity are reasons mentioned for undergoing FGM/C. Marriage between an initiated and an uninitiated member was not possible, and uninitiated women were stigmatized.

Recently HCPs noted a dropping prevalence of FGM/C due to awareness about the negative effects of FGM/C and a change in attitude towards FGM/C and womanhood, as well as the effect of some Christian congregations preaching against it. The Sande society further had trouble finding a *replacement* for passed *zoes* (i.e., traditional circumcizor), which led to a decrease in *bush schools* in Liberia.


*“So, but now most people now when they die, to even get that person is not easy. The young people can run away. Like yesterday one of [the training participants] was sharing with me that somebody that they were supposed to replace that person but because there was no replacement, they carry the body all around until the body turned into pieces.”*
(29, FGD)

Some HCPs positioned themselves against the practice and would like to see it abolished completely, while others would like to *transform* the practice into another symbolic practice, such as pricking or nicking the tip of the clitoris. The symbolic practice might act as a substitute for FGM/C that could help some Sande members to symbolize a satisfactory conclusion of the initiation ceremony. Most HCPs understood the negative health effects of FGM/C. A male nurse, however, warned that *deeply traditional nurses* might still be in favor of continuing FGM/C.


*“I am saying that that practice that those people are doing, the FGM, it can be transformed into another practice that will not be harmful.”*
(14, SSI)

#### 3.3.2. Possible Effect of an FGM/C-‘app’

HCPs believed that health education about the negative health effects of FGM/C through the ‘app’ might help to gradually reduce the dropping prevalence of FGM/C. Patients might act as multiplicators and share the messages received with their family and friends, carrying it into schools, churches and their communities. However, HCPs pointed out that *zoes and traditional leaders*, decision-makers from all regions and religious leaders would need to be trained on FGM/C, as they have influence over the community.

“*We have to get community leaders or stakeholders, involve them in similar gatherings, where they themselves will be able to discuss the issue in the same way we all was discussing. And you take people from almost the 15 subdivisions [counties] to get their views.”*(43, FGD)

Several male HCPs explained that to *succeed in reducing maternal mortality*, a law was needed to address the *issue of FGM/C*. Legislation against FGM/C is thought to enable teaching about FGM/C at school. HCPs emphasized that—though necessary—political commitment against FGM/C might be weak due to FGM/C being deeply rooted in Liberian culture. Some HCPs argued that societal change, in this case abstinence from FGM/C, would be a *gradual process*.

## 4. Discussion

The perceptions of HCPs on a novel mLearning application on care to FGM/C survivors were explored in the frame of a qualitative study carried out in 2019 in Liberia. As per their narratives, the ‘app’, designed to support self-learning, decision-making and follow-up of survivors’ health, was acceptable, feasible and useful. The innovation could be useful to broaden Liberian HCPs' knowledge and to educate their patients and their communities. Such innovations were thought to lead to a better care for FGM/C survivors and to increase visibility about the negative health consequences of FMG/C, possibly contributing to a reduction of FGM/C prevalence.

The findings are aligned with previous research indicating that mLearning tools are efficient in improving HCPs' skills and knowledge [15,23]. Other mLearning interventions in the form of apps have been found acceptable for the sub-Saharan setting [24,25]. A Liberian study showed the potential of mLearning tools being recognized as continuous professional development by the Liberian Board for Nursing and Midwifery and their uptake by Liberian HCPs to improve their clinical practice [26]. To facilitate uptake, users need to fully comprehend the tools, for which participants in our study have suggested trainings [23].

Smartphone ownership and internet connectivity is on a steady rise in sub-Saharan countries, indicating that mLearning programs are feasible [27]. However, as demonstrated in our study, mLearning tools depend on access to mobile phones and connectivity. An off-line application that can be shared free of charge between HCPs is among the solutions suggested by this study participants to overcome this barrier.

Ag Ahmed et al. identified user acceptance as a key success factor for mobile health (mHealth) uptake in sub-Saharan Africa [28]. Co-design strategies can help to increase acceptance and tailor the tool according to the HCPs' needs in a culturally accepted way [29,30]. Especially concerning a deeply entrenched practice, such as FGM/C, the content and presentation of the application needs to be culturally sensitive and adapted to the local circumstances [13,31,32]. In this regard, our participants suggested including culturally appropriate education about FGM/C and media in the form of pictures and videos, as well as an appropriate choice of words.

Consistent with other studies, HCPs have received little education on the health consequences of FGM/C and call for its inclusion in (para)medical academic curricula to provide better care to FGM/C survivors [10,33]. Continuous education, e.g., in the form of case studies and narratives of FGM/C survivors, need to be provided to HCPs so that FGM/C-caused health consequences are no longer perceived as ‘normal’.

Another limiting factor for patient care is the secrecy around FGM/C, which was shown to hamper healthcare-seeking behavior and impedes open dialogue between HCPs and patients [7]. HCP education and FGM/C care need to be supported by the Liberian Ministry of Health to provide a safe environment for HCPs to begin the conversation about FGM/C with patients. Furthermore, psychosocial support should be offered to all FGM/C survivors. The novel FGM ‘app’ might be used by HCPs to discuss the topic and—after successful implementation of the results of this paper and reiteration in a piloting study—promote improved care to FGM/C survivors and health education to patients.

Some of the participants believed that mLearning tools such as the ‘app’ could contribute towards the eradication of the practice of FGM/C in Liberia. In line with this finding, most participants of the workshops perceived their role in advocating against FGM/C, a practice that most deemed harmful for women’s health and viewed as a human rights violation in a questionnaire. To achieve a reduction in the prevalence of FGM/C, HCPs and patient education might contribute but is not the sole driving factor. The teaching activities of HCPs need to be backed up by laws prohibiting FGM/C and their enforcement, as well as the involvement of traditional and religious leaders. In February 2022, the first steps in this direction were taken, as the Liberian government and the National Council of Chiefs and Elders of Liberia declared a three-year moratorium on FGM/C [34]. Once a political discourse about FGM/C commences, a social rethinking and discourse can begin, and education about FGM/C might begin in paramedical curricula, postgraduate education and health and education spaces other than college and university classrooms.

A strength of this study was to involve HCPs of several professions, from trained traditional midwives to physician assistants, male and female, obtaining a broad range of viewpoints. As this research was not a performance usability but a perceptions study, the utility and usability perceived by the participants and the a priori acceptability of the tool is portrayed in this paper. Further research should be developed about piloting with different demo versions and participants’ perception after the sustained use of the ‘app’. Our results are further limited, as not all HCPs had access to a mobile phone and might not be familiar with the use of apps. The impact of the ‘app’ on the prevalence of FGM/C could not be measured, and the opinions about the mLearning tool portrayed might not be generalizable. Further, since the same researcher (K.N.) was present at the workshops and conducted the interviews, participants may have been subject to social desirability bias.

## 5. Conclusions

HCPs accepted using an mLearning tool for self-study, decision-making and the health education of patients about FGM/C. Such a novel tool can increase HCPs' knowledge and be used as a medium to initiate a conversation about FGM/C with survivors, other HCPs and practicing communities. The ‘app’ might lead to better FGM/C detection and management and raise the visibility of health concerns around FGM/C. For the ‘app’ to contribute to a reduction in the prevalence of FGM/C, further advocacy, health education, community mobilization and capacity-building efforts backed by the Liberian government beyond the three-year moratorium are necessary.

## Figures and Tables

**Table 1 ijerph-19-10855-t001:** World Health Organization classification of female genital mutilation/cutting.

Type	Description
I	Partial or total removal of the clitoral glans (clitoridectomy) and/or the prepuce
II	Excision: Partial or total removal of the clitoral glans and the labia minora, with/without excision of the labia majora
III	Infibulation: Narrowing the vaginal opening, with the creation of a covering seal by cutting and appositioning the labia minora or labia majora with/without the excision of the clitoral prepuce and glans
IV	All other harmful procedures to the female genitalia for non-medical purposes, for example, pricking, piercing, incising, scraping and cauterization

Note: Adapted from “Female genital mutilation” by World Health Organization. In Female genital mutilation: Evidence brief. World Health Organization: 2019, p. 1 [1].

**Table 2 ijerph-19-10855-t002:** Socio-demographic characteristics of study participants disaggregated by sex.

Characteristic	Entire Sample *n* = 42	Female *n* = 27	Male *n* = 15
**Age (years) mean ± standard deviation**	39.3 ± 1.4 ^a^	42.2 ± 8.5 ^b^	34.5 ± 6.6
**Occupation**			
Midwife	7 (16.7%) ^c^	7 ^c^	0
Nurse	17 (39.5%) ^c^	8 ^c^	9
Physician assistant	6 (14.3%)	0	6
Trained traditional midwife	12 (28.6%)	12	0
Other	2 (4.8%)	2	0
**Country of Birth**			
Liberia	42 (100.0%)	27	15
**County of Birth**			
Bong	4 (9.5%)	3	1
Lofa	4 (9.5%)	2	2
Margibi	3 (7.1%)	2	1
Montserrado	8 (19.0%)	5	3
Nimba	15 (35.7%)	12	3
Other	8 (19.0%)	3	5
**Marital status**			
Single	4 (9.5%)	0	4
Not-married, cohabitating	14 (33.3%)	6	8
Married	21 (50.0%)	19	2
Other	3 (7.2%)	2	1
**Highest education completed**			
No education	3 (7.1%)	3	0
Primary	4 (9.5%)	4	0
Secondary	6 (14.3%)	6	0
College	14 (33.3%)	7	7
University	15 (35.7%)	7	8
**Religion**			
Christianity	42 (100.0%)	27	15
Baptist	3 (7.1%)	1	2
Catholic	7 (16.7%)	6	1
Jehovah’s Witness	3 (7.1%)	1	2
Lutheran	3 (7.1%)	2	1
Methodist	6 (14.3%)	3	3
Pentecostal	6 (14.3%)	2	4
Other Christian Religion	14 (33.3%)	12	2
**Workshop participant**			
Yes	36 (85.7%)	24	12
No	6 (14.3%)	3	3

^a^ *n* = 41, ^b^
*n* = 26, ^c^ two respondents were both nurse and midwife.

**Table 3 ijerph-19-10855-t003:** Reported mobile phone expertise and the use of mobile phones by study participants disaggregated by sex.

Variable	Entire Sample *n* = 42	Female *n* = 27	Male *n* = 15
**Phone company**			
Itel	12 (28.6%)	7	5
Samsung	4 (9.5%)	3	1
Tecno	14 (33.3%)	9	5
Other	10 (23.8%)	7	3
None	2 (4.8%)	1	1
**Operating System**			
Android	29 (69.0%)	27	12
Feature phone	11 (26.2%)	9	2
No phone	2 (4.8%)	1	1
**Network**			
Lonestar	3 (7.1%)	2	1
Orange	5 (11.9%)	3	2
Both	34 (81.0%)	22	12
**Years of phone usage (mean ± standard deviation)**	11.0 ± 0.8 ^a^	10.7 ± 5.2	11.7 ± 3.8 ^b^
**Average time on the phone per day**			
Mainly for calls	11 (26.2%)	10	1
Less than 2 h	13 (31.0%)	9	4
2–4 h	10 (23.8%)	5	5
4–6 h	3 (7.1%)	1	2
More than 6 h	5 (11.9%)	2	3
**Usage of clinical apps**			
Yes	24 (57.1%)	12	12
No	18 (42.9%)	15	3

^a^*n* = 41, ^b^
*n* = 14.

## Data Availability

The data that supported this study cannot be made available as per research ethics explained to participants during the informed consent process.

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
