# Peer review of "Acceptability of a Smartphone Application to Enhance Healthcare to Female Genital Mutilation Survivors in Liberia: A Qualitative Study"

_ijerph, 2022, doi:10.3390/ijerph191710855_

Round 1

Reviewer 1 Report

Thank you for allowing me to review this article.  Overall, it was well-written, there are only a few places where it did not read as smoothly.  

line 40- secret is in the same sentence twice consider using a different word or synonym for secret.  

line 46-47-The read of this sentence is odd, try survivors benefit from

line206-207- It is unclear what exactly is meant by "respect of African culture" and how does mentioning other places that still have FGM showing respect?  I just could not think of an example while reading this and it is unclear.  

Line390-391-could you describe or further elucidate what is meant by another non harmful practice?  What type of non harmful practice?  like a labia or clitoral piercing?  I think an aexample is needed to further help readers understand

link #4 in the references is not found, please check.

I would recommend that the authors consider their potential audience and provide a bit more context in the introduction.  Some may not know what the Sande Society is a brief explanation of who , what, when , where and the power they hold is important context for the reader.  I have done lectures on FGM/C but this is my first time hearing about the Sande society.  I searched it. 

Please consider providing a bit more explanation of the Maputo Protocol- and why it might be important and lend context to some readers.    

It would be helpful to readers as well if there was a brief and basic explanation of FGM/C and the levels/types.  Although we don't see any discussion about this until page 10, line 375.  It might be helpful to the reader and provide another aspect of important contextual information which should come earlier to help readers understand the particulars for  Liberia.  

There are some statements in the results that particularly concerning and disturbing that are communicated but then not taken up again in the discussion or conclusion, line 178-179, 316-318, subtheme 3.4.1 lines 361-363, line 370-372.  

Consider expanding the conclusion to include a discussion of future directions and anything ongoing.  

Author Response

Manuscript ID: ijerph-1843560, Round 1

Responses to reviewer 1

Reply: We thank you for your extensive and helpful review. We have addressed your observations and suggestions in a point-by-point manner below and highlighted them in the manuscript using track changes.

Thank you for allowing me to review this article. Overall, it was well-written, there are only a few places where it did not read as smoothly.  

Reply: We thank the reviewer for appreciating our work. We appreciate the following comments and changed them as highlighted:

  • line 40- secret is in the same sentence twice consider using a different word or synonym for secret. Reply: We rephrased the sentence.
  • line 46-47-The read of this sentence is odd, try survivors benefit from Reply: We changed the word “of” to “from”.
  • line206-207- It is unclear what exactly is meant by "respect of African culture" and how does mentioning other places that still have FGM showing respect?  I just could not think of an example while reading this and it is unclear. Reply: We added an explanation. The improved phrase reads as follows “Some participants pointed out that the ‘app’ should respect the African culture at any point, i.e., be culturally sensitive when talking about cultural practices and their consequences, and display to its user that efforts to reduce the prevalence of FGM/C are not limited to Liberia.”.
  • Line390-391-could you describe or further elucidate what is meant by another non harmful practice?  What type of non harmful practice?  like a labia or clitoral piercing?  I think an aexample is needed to further help readers understand. Reply: We rephrased the sentence and included examples to facilitate understanding.  
  • link #4 in the references is not found, please check. Reply: We checked all links in the references and confirm their correct functioning.

I would recommend that the authors consider their potential audience and provide a bit more context in the introduction.  Some may not know what the Sande Society is a brief explanation of who, what, when, where and the power they hold is important context for the reader.  I have done lectures on FGM/C but this is my first time hearing about the Sande society.  I searched it. 

Reply: Thank you for pointing this out. We included more information of the Sande society in the introduction.

Please consider providing a bit more explanation of the Maputo Protocol- and why it might be important and lend context to some readers. 

Reply: We thank the reviewer for this comment and added an explanation about the Maputo Protocol.

It would be helpful to readers as well if there was a brief and basic explanation of FGM/C and the levels/types.  Although we don't see any discussion about this until page 10, line 375.  It might be helpful to the reader and provide another aspect of important contextual information which should come earlier to help readers understand the particulars for Liberia.  

Reply: Thank you for this valuable suggestion. We implemented a table with the description of the FMG/C types in the introduction and expanded the explanation about FGM/C.

There are some statements in the results that particularly concerning and disturbing that are communicated but then not taken up again in the discussion or conclusion, line 178-179, 316-318, subtheme 3.4.1 lines 361-363, line 370-372.  

Reply: We thank the reviewer for this suggestion and addressed all mentioned statements in the corresponding parts of the discussion.

Consider expanding the conclusion to include a discussion of future directions and anything ongoing.  

Reply: We appreciate this suggestion and expanded the conclusion.

With the changes made we hope that the manuscript will be suitable for publication in IJERPH.

Kim Nordmann

Reviewer 2 Report

I congratulate you on the subject on which you focus your study. It can be improved in:

RESULTS

In qualitative methodology, the profile of the people participating in a study is reflected in the Methodology section, under Participants. I suggest to the authors that the information on lines 143-150 including Table 1.

The text on lines 82-94 does not correspond to the heading. This text speaks of the procedure not the sample. 

BIBLIOGRAPHIC REFERENCES

No. 13 is not well written. 

Author Response

Manuscript ID: ijerph-1843560, Round 1

Responses to reviewer 2

I congratulate you on the subject on which you focus your study.

Reply: We thank the reviewer for appreciating the focus of our study.

It can be improved in:

RESULTS

In qualitative methodology, the profile of the people participating in a study is reflected in the Methodology section, under Participants. I suggest to the authors that the information on lines 143-150 including Table 1.

Reply: We appreciate this suggestion and included this section under the heading “Population and sampling procedure” and “Data collection and analysis”.

The text on lines 82-94 does not correspond to the heading. This text speaks of the procedure not the sample. 

Reply: We thank the reviewer for this comment and changed the heading to “Population and sampling procedure”.

BIBLIOGRAPHIC REFERENCES

No. 13 is not well written. 

Reply: We thank the reviewer for pointing this out and updated reference No. 13.

With the changes made we hope that the manuscript will be suitable for publication in IJERPH.

Kim Nordmann